# Electrodeposition of Soft Magnetic Fe-W-P Alloy Coatings from an Acidic Electrolyte

Natalia Kovalska [1,2,3,*], Antonio Mulone [4], Jordi Sort [5,6], Uta Klement [4], Gurdial Blugan [3], Wolfgang Hansal [2] and Wolfgang Kautek [1,*]

1  Department of Physical Chemistry, University of Vienna, A-1090 Vienna, Austria
2  RENA Technologies Austria GmbH, A-2700 Wiener Neustadt, Austria; wolfgang.hansal@elektrochemie.eu
3  Laboratory for High Performance Ceramics, Empa, Swiss Federal Laboratories for Materials Science & Technology, CH-8600 Dübendorf, Switzerland
4  Department of Industrial and Materials Science, Chalmers University of Technology, SE-41296 Gothenburg, Sweden; mulone@chalmers.se (A.M.); uta.klement@chalmers.se (U.K.)
5  Department de Física, Facultat de Ciències, Universitat Autònoma de Barcelona, E-08193 Bellaterra, Spain; sort.jordi@gmail.com
6  Institució Catalana de Recerca i Estudis Avançats (ICREA), E-08010 Barcelona, Spain
*  Correspondence: natalia.kovalska@empa.ch (N.K.); wolfgang.kautek@univie.ac.at (W.K.)

**Abstract:** Fe-W-P coatings were deposited from a newly developed electrolytic bath. The effect of plating parameters, such as electrolyte current density and pH has been studied. It was found that the pH has a very strong effect on the phosphorous content of the coatings. Metallic-like, non-powdery alloys of Fe-W-P deposits with no cracks (lowly stressed) can be obtained at a lower pH (<3), exhibiting high phosphorus (up to 13 at.%) and low tungsten (6 at.%) contents. At a higher pH (>3), the composition changes to low phosphorus and high tungsten content, showing a matte, greyish, and rough surface. The applied current density also influences the morphology and the amount of phosphorous content. The deposits showed an amorphous structure for all samples with soft ferromagnetic properties.

**Keywords:** electrodeposition; Fe-W-P alloy; hydrogen evolution; pH; soft magnetic





## 1. Introduction

The current trends towards the reduction of hazardous procedures and cost competitiveness become dominant in many new applications such as in micro-electromechanical systems (MEMS) and nano-electromechanical systems (NEMS). Amorphous iron group coatings with phosphorous or/and tungsten are of industrial interest due to their soft magnetic behaviour, application in recording media, sensors, data storage media, inductive devices, and transformer cores [1–6]. It has been shown that the addition of phosphorous can improve the magnetic properties of electrodeposited coatings [7,8]. A detailed study based on electrochemical in-situ techniques led to a mechanistic understanding of the Fe-P alloy electrodeposition [9]. An increase of the P content exerts a beneficial effect on the corrosion resistance [10,11].

The addition of P in Co-W and Fe-W coatings results in compact and dense layers with decreased grain size [12–16]. The presence of W in Fe-P alloys significantly improves properties such as hardness, heat stability, wear resistance, corrosion protection, and magnetic characteristics [15,17–20]. This Fe-W-P coating can compete with the high-cost Ni-W-P alloy used in decorative finishing, and can be considered as an alternative to Cr, which is produced in the hazardous process based on Cr(VI) [21].

The electrodeposition of W can only take place in the presence of iron group metals through an induced co-deposition [13,22]. Most Fe-W-P alloys studied so far exhibit the presence of a higher amount of W and a low P content. These coatings are generated

from electrolytes containing $Na_2WO_4 \cdot 2H_2O$, as a W source at a moderate alkaline pH (~8). Some authors obtained Fe-W alloys by means of electrodeposition in citrate-containing solutions [7,8,17,18,22]. A generally accepted mechanism for induced co-deposition of W with iron group metals (M) in the presence of citrate implies the formation of M(II)-citrate complexes, which favours tungstate reduction to adsorbed species composed of W oxides with citrate complexes [21,22].

Considering the numerous advantages of Fe-P coatings with high P content [9–11], the aim of this work is to explore a new electrolytic bath using sustainable or low quantity hazardous materials, which leads to an increase in the P content in Fe-W-P alloys. Previous studies reported the formation of alloys, where the P content dominates over W. Therefore, we describe the use of a new source of elements and the influence of the chemical environment on the final alloy composition.

## 2. Materials and Methods

The electrolyte components for Fe-W-P electrodeposition are shown in Table 1. Analytical grade chemicals and deionized water were used. The pH was adjusted to 3 and 4 by the addition of $H_2SO_4$ and NaOH. All electrodeposition experiments were carried out at 60 °C.

**Table 1.** Bath composition.

| Components | Molarity/M | Concentration/gL$^{-1}$ |
| --- | --- | --- |
| $FeSO_4 \cdot 7H_2O$ | 0.72 | 200 |
| $H_2PO_4 \cdot 12WO_3 \cdot H_2O$ | 0.02 | 60 |
| Citrate | 0.39 | 100 |
| Glycine | 0.64 | 48 |
| $NaH_2PO_2 \cdot H_2O$ | 0.08 | 7 |
| $(NH_4)_2SO_4$ | 0.45 | 60 |

The Electrochemical Quartz Microbalance (EQMB) measurements were performed with a three-electrode system equipped with an EQCM module (Autolab PGSTAT, Metrohm, Filderstadt, Germany). 6 MHz AT-cut quartz crystals were coated with 100 nm thick gold layers. An area of 0.384 cm$^2$ was exposed. A gold coil was used as a counter electrode and a Ag/AgCl/3M KCl electrode was used as a reference electrode (+0.21 V vs. standard hydrogen electrode, SHE) [23]. For easier data evaluation the potentials are presented versus the SHE as $U_{SHE}$. The gold surface was cleaned by cycling in 0.1 M $H_2SO_4$ solution.

Galvanostatic deposition experiments were performed in a beaker of 0.3 l without stirring. A brass plate served as a substrate. All samples were rinsed in 0.1 M $H_2SO_4$ before deposition. The thickness of the electrodeposited coatings varied between 11 and 15 μm. The thickness was calculated based on weight elemental analyses of the electrodeposited alloys [24]. The reference electrode was Ag/AgCl/3M KCl (+0.21 V vs. standard hydrogen electrode, SHE) [23]. As the counter electrode served a steel plate (SAE 304). It exposed ~30 cm$^2$ to the electrolyte and was positioned parallel to the working electrode at a distance of ~10 cm. The current efficiency of the alloy deposition was determined by Faraday's law [25].

The coating surfaces were characterized with a scanning electron microscope (Hitachi FEG-SEM S4800, Tokyo, Japan) equipped with EDX for elemental analysis. An acceleration voltage of 5 kV was used for the imaging mode, and a voltage of 20 kV for the EDX mode. The vacuum pressure in the specimen chamber was of the order of $10^{-4}$ Pa. Crystallographic information was obtained by means of an X-ray diffractometer (Bruker AXS D8, Karlsruhe, Germany) operated with Cr Kα radiation (λ = 2.0821 Å) at 35 kV and 40 mA, respectively.

Hysteresis loops were performed at room temperature using a vibrating sample magnetometer (EZ9 Microsense, LOT Quantum Design, Darmstadt, Germany) with a

maximum magnetic field of 20 kOe. The loops were measured applying the field along the in-plane direction of the films.

## 3. Results and Discussions

The development of the presently investigated Fe-W-P electrolytic bath was based on electrolytes for the deposition of semi-amorphous, smooth, porous free Fe-P alloys [11]. To obtain a stable Fe-W-P electrolyte for long-term use, an acidic solution with the tungsten source phosphotungstic acid (PTA; $H_2PO_4 \cdot 12WO_3 \cdot H_2O$) was chosen in the present study, which had not been used before. The conjugate base of this heteropoly acid is the oxyanion $PW_{12}O_{40}^{3-}$, where W has an oxidation state of +6 [26]. PTA decomposes with increasing the pH [26,27].

### 3.1. Electrochemical Quartz Microbalance

The Fe-W-P alloy co-deposition was studied by the electrochemical quartz microbalance (EQMB). The measurements were performed to evaluate the influence of pH on the Fe-W-P deposition, compared to the Fe-W alloy deposition (Figure 1). The standard potentials of the $Fe/Fe^{2+}$ electrode ($U° = -0.44$ V [28]) and the phosphorous deposition either from hypophosphate (oxidation state +1) [29],

$$H_3PO_2 + H^+ + e^- \rightleftharpoons P + H_2O \ (U° = -0.51 \text{ V}), \tag{1}$$

or from phosphotungstic acid (oxidation state +5) [29],

$$H_3PO_4 + 5H^+ + 5e^- \rightleftharpoons P + 4H_2O \ (U° = -0.40 \text{ V}) \tag{2}$$

are close to each other.

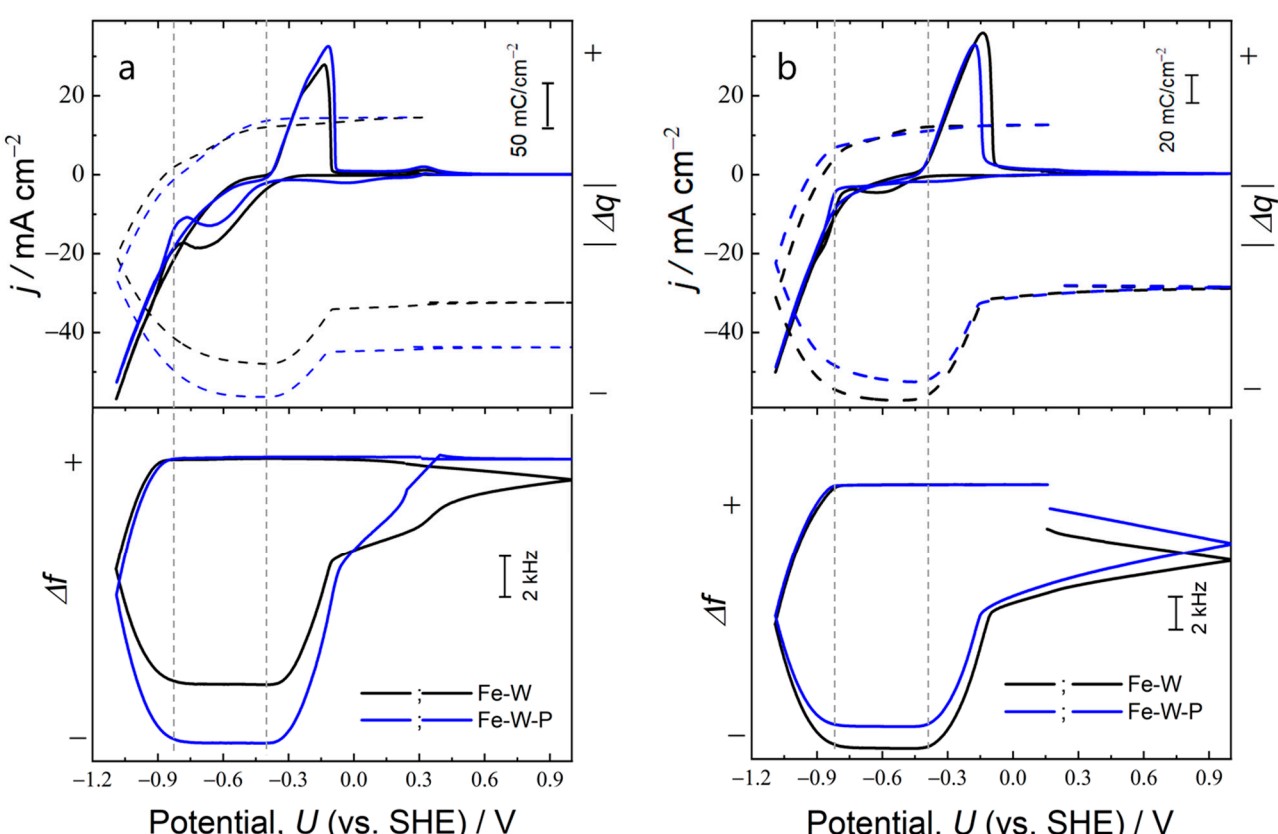

**Figure 1.** Electrochemical quartz microbalance data of the Fe-W and Fe-W-P electrodeposition (third potential cycle): current and charge density, and frequency change. (**a**) pH 3 and (**b**) pH 4.

An EDX analysis of a deposit from an electrolyte without hypophosphate ($NaH_2PO_2 \cdot H_2O$) resulted in practically zero (0.2–0.3 at.%) P content, even though phosphate was present in the form of $H_2PO_4 \cdot 12WO_3 \cdot H_2O$ (comp. Table 1). In this case, the W content indicated 6–8 at %.

A negative potential scan in the Fe-W and the Fe-W-P baths at pH 3 resulted in a negative current wave at ca. −0.4 V (Figure 1a), which can be interpreted as hydrogen evolution supported by the measured charge density $|\Delta q|$ [9]. The absence of any frequency change ($\Delta f$), i.e., any mass increase, indicates the generation of a gas uncoupled from the surface. With a negative potential of −0.85 V, co-deposition of the alloy components is indicated by a mass and current increase. The phosphotungstic acid contains W in the oxidation state of +6 (comp. $WO_3$). Its reduction to elementary tungsten occurs via the following partial steps [30]:

$$WO_3 + 2H^+ + 2e^- \rightleftharpoons WO_2 + H_2O \ (U^\circ = -0.036 \text{ V}) \tag{3}$$

$$WO_2 + 4H^+ + 4e^- \rightleftharpoons W + 2H_2O \ (U^\circ = -0.119 \text{ V}) \tag{4}$$

The respective Nernst potential of the W formation Equation (4) at pH 3 and 4, respectively, is approximately −0.8 V, which is close to the appearance potential of the Fe−W-P alloy (Figure 1). The hydrogen evolution is enhanced by lowering the pH, as seen by the increase of the *j*-wave and the charge increase negative of ca. −0.40 V.

The anodic stripping peaks of the alloys exhibit an asymmetry indicating the formation of a passive layer, most probably due to tungsten oxides, so that the potentiostat cannot compensate for the IR drop between the working and the reference electrodes. This phenomenon has not been observed without W [9]. After the abrupt decay of the anodic stripping peak, approximately 30–40% of the alloy mass remained. In the course of the further anodic potential scan this mass is further reduced practically without any charge transfer due to the chemical dissolution (hydrolysis) of the passive W phases [31], e.g.,

$$WO_{3(s)} + H_2O \rightarrow H_2WO_{4(aq)} \tag{5}$$

or

$$WO_{3(s)} + OH^- \rightarrow WO_2{}^{4-}{}_{(aq)} + H^+ \tag{6}$$

### 3.2. Influence of pH, P and W on the Coatings

A composition analysis was performed on the electrodeposited Fe-W-P coatings. The P and W content varies by changing the pH, as shown in Figure 2a. The current density shows a minor influence except in the case of P at low *j*. The reduction of pH from 4 to 3 leads to an increase in P and W contents [32]. At a higher $H^+$ concentration, the solubility of $H_3PO_{2(aq)}$ and $H_3PO_{4(aq)}$ species is higher [33] supporting a higher P deposition rate. Moreover, an increase in $H^+$ supports the reduction reaction of both P and W according to Equations (1), (3) and (4). The current efficiency ($\eta$) is higher in a less acidic electrolyte, i.e., at pH 4, and is more or less unaffected by the change in current density (Figure 2b). This can be explained by a lower hydrogen evolution competing with the alloy deposition (comp. Figure 1b). In case of an electrolytic bath at pH 3, the current efficiency increases with higher current density [32]. The lowest current efficiency is observed at a current density of 30 mA cm$^{-2}$ and pH 3 (Figure 2b) resulting in a slow deposition rate of the alloy and a maximum P content of ca. 13 at.% (Figure 2a).

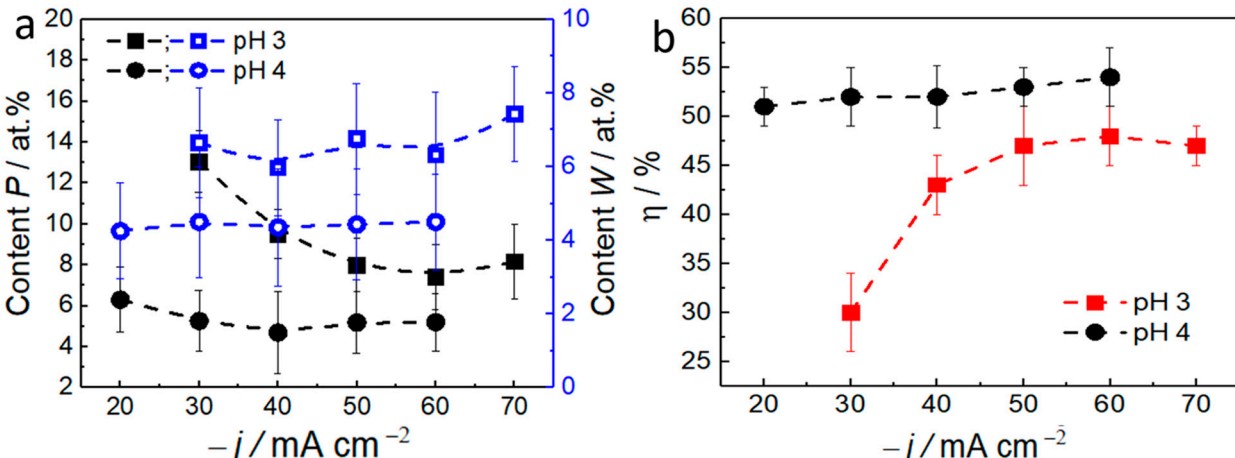

**Figure 2.** (**a**) Dependence of the P (black and solid) and W (blue and open) content in Fe-W-P coatings on the electrodeposition current density *j* at two acidic pH values. (**b**) Dependence of the current efficiency η on the electrodeposition current density *j* of the Fe-W-P electrodeposition at two acidic pH values.

The SEM images of the alloy show the effect of the pH and current density (Figure 3). The observed layer morphology is controlled by the competition between hydrogen bubble evolution, bubble detachment, and the layer growth. pH and *j* control η [32] (Figure 2b). A low η value indicates a high rate of $H^+$ reduction, a vigorous hydrogen gas evolution, and therefore a fast bubble detachment [34].

At pH 4, η is relatively high and practically independent of *j*. The $H_2$-bubble formation and detachment are relatively slow, so that an overgrowth of the rapidly depositing Fe-W-P alloy leads to partially cracked film bumps which appear hollow (Figure 3). A powdery dark greyish coating character is observed by the naked eye. Additionally, the SEM resolution is limited due to charging of the low conductivity deposit. This phase with low P content (Figure 2) exhibits an enhanced corrosion/oxidation rate [10,11] so that its conductivity drops.

At pH3, the bumps are not cracked. η is relatively low and the bubble formation and detachment is extremely vigorous. Therefore, the overgrowth of attached $H_2$-bubbles could not take place. The good resolution of the SEM images at all magnifications indicates low charging, suggesting a high conductivity of the coating with an obviously low oxidic conversion overlayer. The naked eye aspect is glossy and not powdery. For this reason, this coating type (pH3) is further investigated in the following.

The elements' partial current distribution, $j_e$, of the corresponding element "e", i.e., Fe, W, and P, were determined at pH 3 (Figure 4). The partial current densities were estimated from the EDX data and the weighed coating mass [11]. $j_e$ as a function of the overall current density, shows that the current efficiency η of electrodeposition was increasing with *j*. $j_e$ of the hydrogen evolution (side reaction) is much less affected by *j* than all other alloy components [34]. The exceptionally high P content at 30 mA cm$^{-2}$ and pH3 (Figure 2a) can be understood on the basis of the evaluation in Figure 4. The relative deposition rate of Fe is much lower than that of P and W. At *j* > 30 mA cm$^{-2}$, all elements show a practically similar deposition rate increase with *j*.

| pH | $-j/\text{mA cm}^{-2}$ | |
| :---: | :---: | :---: |
| | **30** | **50** |

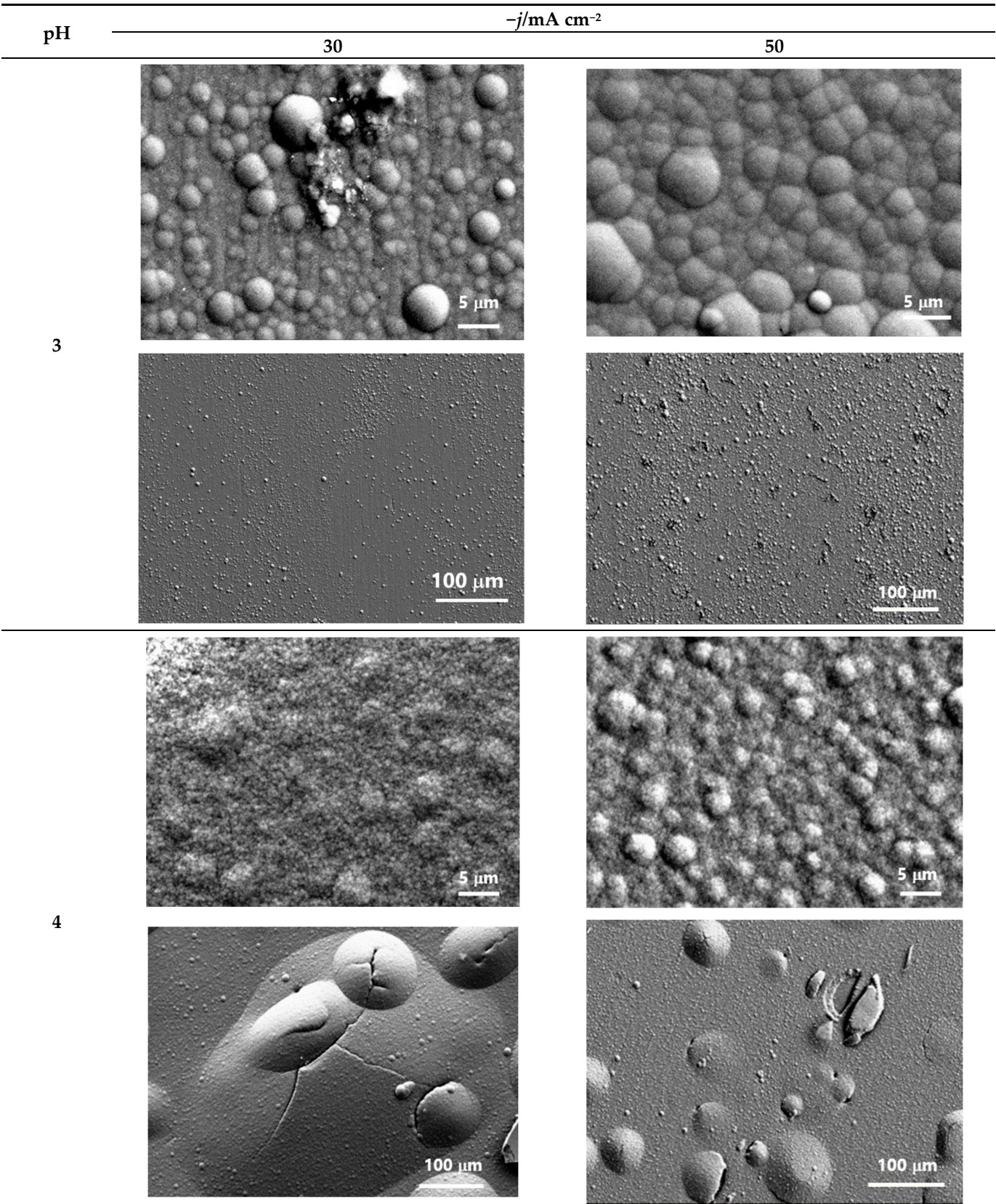

**Figure 3.** SEM morphology of Fe-W-P coatings deposited with DC electrodeposition at pH 3 and 4, respectively, and current density 30 and 50 mA cm$^{-2}$. Images presented with different magnifications.

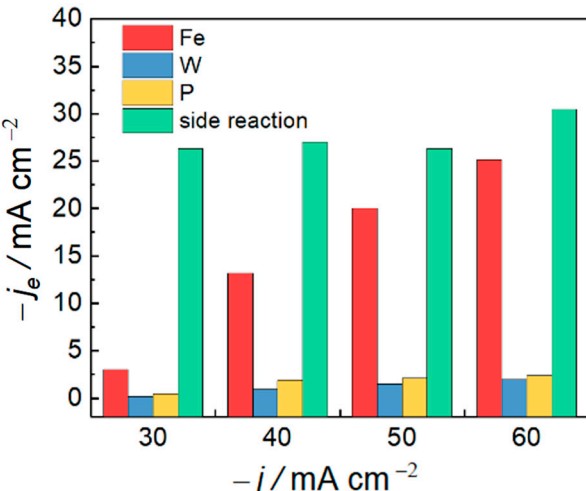

**Figure 4.** Partial deposition current densities $j_e$ of the elements Fe, W, P, and the hydrogen evolution (side reaction), $j$ vs. the applied current densities. pH 3.

XRD results of the Fe-W-P coatings are presented in Figure 5. The three samples show an amorphous structure, as indicated by the presence of a broad peak from 65° to 75°. This peak appears towards lower angles with respect to the $2\theta$ (110) reflection characteristic of bcc-iron (i.e., 68.78°). This is due to the W substitution in the bcc-iron lattice, resulting in an increased lattice parameter [20]. The only visible crystalline peak corresponds to the brass substrate. The varying intensities of the brass peak for the various at.% P are due to the respective coating thicknesses.

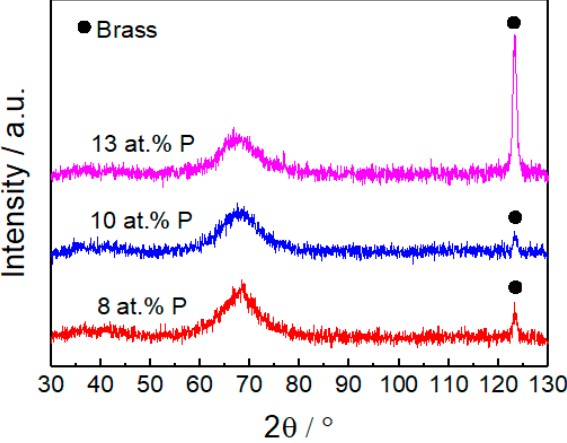

**Figure 5.** X-ray diffractograms of the Fe-W-P coatings deposited with increasing P content.

*3.3. Magnetic Properties*

Previous studies showed that the amounts of W and P in this type of alloy have an influence on the magnetic properties of the coatings [18,35–37]. In particular, the coatings are soft magnetic (with almost negligible coercivity) when the W or P contents are beyond 20 at.%. In the present study, magnetic measurements were performed applying the magnetic field in the plane of the films (Figure 6). The obtained results reveal that the Fe-W-P alloys are soft magnetic with coercively values lower than 25 Oe, although the percentages of the non-metallic element P in the alloy compositions were lower than 20 at.%. This is probably due to a negligible magnetocrystalline anisotropy, as was reported for other amorphous or nanocrystalline alloys containing a non-ferromagnetic element [38,39]. The addition of non-metallic elements P to Fe and W also leads to a reduction of the saturation magnetization, $M_s$ [35].

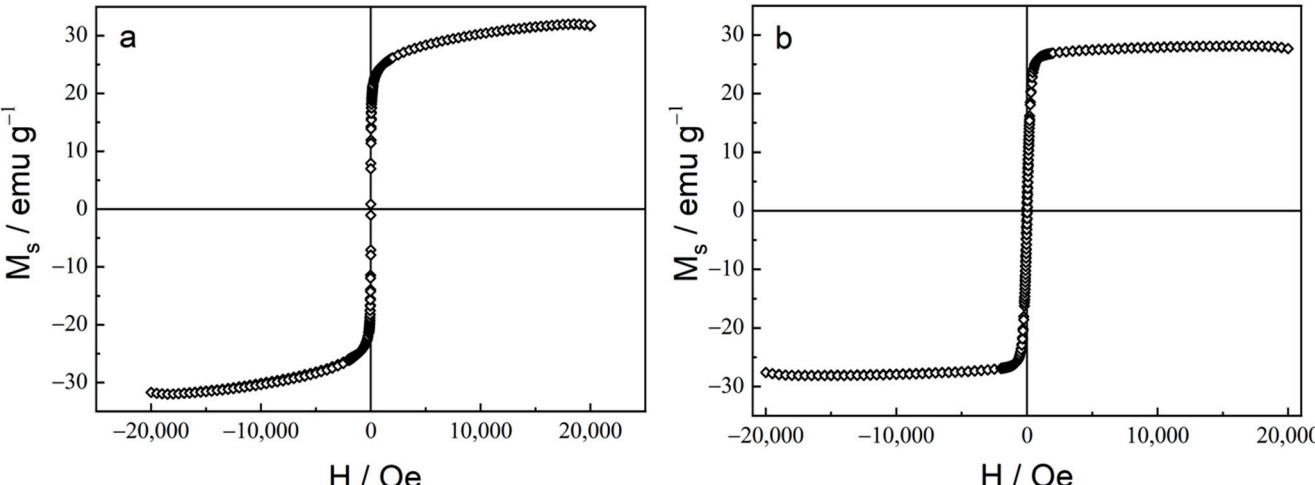

**Figure 6.** Hysteresis loops of Fe-W-P coatings. (**a**) 8 at.% P, 6 at.% W; (**b**) 13 at.% P, 6 at.% W, recorded with an external magnetic field applied along the in-plane direction.

For example, Figure 6 compares two alloys with the same amount of W (6 at.%) and different P contents. All deposited samples exhibit low ferromagnetic signals, i.e., $M_s$ = 20−40 emug$^{-1}$. The reason for such low $M_s$ could be that a mixture of non-ferromagnetic and weakly ferromagnetic regions co-exist in the coatings [35]. The increase of the P content significantly changes the shape of the loop (making it more square-like) and increases coercivity (i.e., from 3 to 24 Oe, for P contents of 8 and 13 at %, respectively).

## 4. Conclusions

A novel sustainable electrolytic bath has been developed yielding a higher content of P in amorphous smooth, porous-free Fe-W-P alloys. The stable electrolyte consists of an acidic solution with the W source phosphotungstic acid. The Fe-W-P alloy co-deposition was studied by the electrochemical quartz microbalance (EQMB) and EDX analysis. Hydrogen evolution takes place at a negative value of less than −0.4 V (SHE) at a pH of 3 and/or 4, indicating a moderate overpotential vs. the Nernst potential. For a negative potential of −0.85 V co-deposition of the alloy components was evident from a mass and current increase. The overpotential of the W deposition was low (~−0.05 V) in contrast to that during the Fe-P alloy deposition. A reduction of pH leads to an increase in P and W. At higher H$^+$ concentration, the solubility of H$_3$PO$_{2(aq)}$ and H$_3$PO$_{4(aq)}$ species are higher and an increase in deposition rate is observed. The current efficiency increases at pH 4 and is mostly unaffected by the overall current density, due to lower hydrogen evolution that competes with the alloy deposition. The layer morphology was controlled by the competition between hydrogen bubble evolution, their detachment, and the layer growth. The partial current $j_e$ distribution of the corresponding element, i.e., Fe, W, and P, were determined at pH 3 from the EDX data and the weighed coating mass. The current efficiency of alloy electrodeposition increases with current density and, as the partial current of elements. The partial current of the hydrogen evolution is less affected by the current density than all other alloy components. The XRD results show that the Fe-W-P alloys are characterized by an amorphous structure. The alloys exhibit relatively low ferromagnetic signals between $M_s$ = 20 and 40 emu g$^{-1}$ of all measured coatings, possibly due to a mixture of non-ferromagnetic and weakly ferromagnetic phases. The coercively values are typical for a soft magnetic material.

**Author Contributions:** Conceptualization: N.K., W.H. and W.K.; Methodology: N.K. and W.K.; Validation: N.K., A.M., U.K., J.S., W.H., G.B. and W.K.; Formal Analysis: N.K. and W.K.; Investigation: N.K. and W.K.; Resources: W.H.; Writing—Original Draft Preparation: N.K.; Writing—Review and Editing: N.K., W.K., J.S. and U.K.; Visualization: N.K. and W.K.; Supervision: W.K.; Project

Administration: W.H.; Funding Acquisition: W.H. All authors have read and agreed to the published version of the manuscript.

**Funding:** The authors acknowledge funding by the HORIZON2020 SELECTA project (No. 642642), the Spanish Government (PID2020-116844RB-C21 and associated FEDER), and the Generalitat de Catalunya (2017-SGR-292).

**Institutional Review Board Statement:** Not applicable.

**Informed Consent Statement:** Not applicable.

**Data Availability Statement:** The data presented in this study are available on request from the corresponding author.

**Acknowledgments:** Experimental assistance by V. Gman (RENA Technologies Austria GmbH), G. Quorri (RENA Technologies Austria GmbH), and R. Mann (RENA Technologies Austria GmbH) are gratefully acknowledged.

**Conflicts of Interest:** The authors declare no conflict of interest. The funders had no role in the design of the study; in the collection, analyses, or interpretation of data; in the writing of the manuscript; or in the decision to publish the results.

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
