# Peer review of "Electrodeposition of Soft Magnetic Fe-W-P Alloy Coatings from an Acidic Electrolyte"

_coatings, doi:10.3390/coatings13040801_

Round 1

Reviewer 1 Report

This manuscript focuses on the synthesis of Fe-W-P micrometre-range films by electrodeposition. The study explores two acidic pH levels and the electrocrystallisation of Fe-W and Fe-W-P. I believe the study fits the journal’s scope and is well written overall. However, I have some questions and suggestions to the authors, as follows:

1-     Line 54, what makes this electrolyte sustainable? How does it differentiate from the state of the art? The answers to these questions should be clearly indicated.

2-     Line 65, how was the gold film deposited? It should be indicated.

3-     Line 75, what steel was used? It should also be indicated.

4-     Line 90, do authors mean ‘base’ instead of ‘basis’?

5-     Line 107, I suggest presenting a table containing the semi-quantitative EDX readings of the films obtained from both electrolytes (with and without hypophosphate).

6-     Figure 1, how was charge density value obtained? You do not refer to this parameter anywhere else in the manuscript. What is its relevance and what information can be obtained from it?

7-     Lines 133-134, did you consider brass substrate dissolution? Any comments on that?

8-     Line 150, it should be said that this is only valid for pH = 4. Figure 2b clearly shows that at pH = 3 the current efficiency decreases with the current density.

9-     Figure 2, are these j values cathodic? If so, they should formally be negative or they should be explicitly indicated as cathodic.

10-  Line 167, commenting on the resolution, perhaps you should indicate the operation pressure inside SEM chamber, as well as beam energy.

11-  Lines 168-169, it is not clear to me how decreasing the conductivity of the phase, would result, per se, in lower resistance to corrosion/oxidation.

12-  Line 170, how do you know the bumps are hollow? How do they compare to those obtained at pH 4? Perhaps some of the H2 bubbles at pH 3 are just smaller and remain under the film.

13-  Line 173, it is still not clear to me this correlation. Perhaps you could explain a bit better.

14-  Figure 3, line 177, two pH and two current densities sound more adequate than ‘various’ in my opinion.

15-  Figure 4, for easier interpretation and formal representation, the xx axis should be arranged to include only the values of the classes considered: 30, 40, 50, and 60 mA/cm2.

16-  Line 210, why do you consider tungsten as a non-metallic element? Tungsten is a transition metal.

17-  Lines 219-220, it is stated that increasing P content increases coercivity. However, this seems to contradict what is suggested in lines 202-203, that the addition of W and P decreases coercivity. Also, I believe there is a typo in line 219 in ‘coercivity’.

18-  Line 241, where were these values (20 and 40 emu/g) read/obtained from?

19-  References list looks a bit outdated. The most recent papers are from 2019 (only 3/37) and most of the cited works are older than 10 years (21/37). I seriously recommend an update on the references.

Author Response

Comments and correction (Reviewer 1)

Comments and Suggestions for Authors

This manuscript focuses on the synthesis of Fe-W-P micrometre-range films by electrodeposition. The study explores two acidic pH levels and the electrocrystallisation of Fe-W and Fe-W-P. I believe the study fits the journal’s scope and is well written overall. However, I have some questions and suggestions to the authors, as follows:

Line 54, what makes this electrolyte sustainable? How does it differentiate from the state of the art? The answers to these questions should be clearly indicated.

Corrected: (Line 54-59)

Considering the numerous advantages of Fe-P coatings with high P content [9–11], the aim of this work is to explore a new sustainable electrolytic bath using non-hazardous materials, which leads to an increase in the P content in Fe-W-P alloys. Previous studies reported the formation of alloys, where the P content dominates over W. Therefore, we describe the use of a new source of elements and the influence of the chemical environment on the final alloy composition.

Line 65, how was the gold film deposited? It should be indicated.

Comment: We did not deposit the gold coating, it was ordered together with the equipment.

Line 75, what steel was used? It should also be indicated.

Corrected (Line 80):

As counter electrode served a steel plate (type 304) exposing ~ 30 cm² to the electrolyte.

Line 90, do authors mean ‘base’ instead of ‘basis’?

Corrected: (Line 95):

The conjugate base of this heteropoly acid is the oxyanion PW12O403−, where W has an oxidation state of +6.

Line 107, I suggest presenting a table containing the semi-quantitative EDX readings of the films obtained from both electrolytes (with and without hypophosphate).

Corrected: Line 112 (Some numbers were added)

An EDX analysis of a deposit from an electrolyte without hypophosphate (NaH2PO2·H2O) resulted in practically zero (0.2 - 0.3 at.%) P content even though phosphate was present in the form of H2PO4·12WO3·H2O (comp. Table 1). In this case, the W content in the coating indicated 6 – 8 at. %.

Figure 1, how was charge density value obtained? You do not refer to this parameter anywhere else in the manuscript. What is its relevance and what information can be obtained from it?

Corrected: Line 118-119 (the value was obtained by QMB.)

A negative potential scan in the Fe-W and the Fe-W-P baths at pH 3 resulted in a negative current wave at ca. -0.4 V (Figure 1a) which can be interpreted as hydrogen evolution by the measured charge density |∆q| [9]. The absence of any frequency change (Df), i.e. any mass increase, indicates the generation of a gas uncoupled from the surface.

Lines 133-134, did you consider brass substrate dissolution? Any comments on that?

Commented: Line 140

Since  a QMB with gold electrodes was used, no dissolution was observed

Line 150, it should be said that this is only valid for pH = 4. Figure 2b clearly shows that at pH = 3 the current efficiency decreases with the current density.

Corrected: Line 155

The current efficiency (η) is higher in a less acidic electrolyte, i.e. at pH 4 and mostly unaffected by the change in current density (Figure 2b). This can be explained by a lower hydrogen evolution competing with the alloy deposition (comp. Figure 1b). In case of an electrolytic bath at pH 3, The current efficiency increases with higher current density [32] . The lowest current efficiency is observed at a current density of 30 mA cm-2 and pH 3 (Figure 2b) resulting in a slow deposition rate of the alloy and a maximum P content of ca. 13 at.% (Figure 2a).

Figure 2, are these j values cathodic? If so, they should formally be negative or they should be explicitly indicated as cathodic.

Corrected: Figure 2

Line 167, commenting on the resolution, perhaps you should indicate the operation pressure inside SEM chamber, as well as beam energy.

Lines 168-169, it is not clear to me how decreasing the conductivity of the phase, would result, per se, in lower resistance to corrosion/oxidation.

Corrected: The good resolution of the SEM images at all magnifications indicates low charging suggesting a high conductivity of the coating with an obviously low oxidic conversion overlayer.

Line 170, how do you know the bumps are hollow? How do they compare to those obtained at pH 4? Perhaps some of the H2bubbles at pH 3 are just smaller and remain under the film.

Corrected: At pH4, is relatively high and practically independent of j. The H2-bubble formation and detachment are relatively slow, so that an overgrowth of the rapidly depositing Fe-W-P alloy leads to partially cracked film bumps which appear hollow (Figure 3)..

Line 173, it is still not clear to me this correlation. Perhaps you could explain a bit better.

??

Figure 3, line 177, two pH and two current densities sound more adequate than ‘various’ in my opinion.

Corrected: Figure 3. Line 186

Figure 4, for easier interpretation and formal representation, the xx axis should be arranged to include only the values of the classes considered: 30, 40, 50, and 60 mA/cm2.

Line 210, why do you consider tungsten as a non-metallic element? Tungsten is a transition metal.

Corrected: Line 220 „The obtained results reveal that the Fe-W-P alloys are soft magnetic with coercively values lower than 25 Oe, although the percentages of the non-metallic element P in the alloy compositions were lower than 20 at.%.”

Lines 219-220, it is stated that increasing P content increases coercivity. However, this seems to contradict what is suggested in lines 202-203, that the addition of W and P decreases coercivity. Also, I believe there is a typo in line 219 in ‘coercivity’.

Comment: The text above, it is said that addition of P and W reduces MS, but it does not say that it also reduces Hc.

Line 241, where were these values (20 and 40 emu/g) read/obtained from?

Comment: The raw data was not normalized (the units were "emu"). It is common to present data divided by mass. Therefore the signal (in emu) was divided by the mass (in g) to obtain emu/g.

References list looks a bit outdated. The most recent papers are from 2019 (only 3/37) and most of the cited works are older than 10 years (21/37). I seriously recommend an update on the references.

Comment: Updated

Reviewer 2 Report

The authors investigated the effect of pH and current density on the composition and microstructure of soft magnetic Fe-W-P coatings. This is a relevant topic to “Coatings”. The experiments are well designed. And the manuscript is well written with interesting results. I believe it will be a valuable and useful resource to the readers of “Coatings”. I would like to recommend “Accept in present form”. 

Author Response

Dear reviewer, 

We are pleased that our article was suitable in its current form.

Best regards, 

Dr. Natalia Kovalska

Reviewer 3 Report

Coatings-2309976-

Electrodeposition of soft magnetic Fe-W-P alloy coatings from an acidic electrolyte 

In this paper, the authors studied the Fe-W-P coatings galvanostatic deposited from a newly developed electrolytic bath. The effect of plating parameters such as electrolyte current density and pH has been investigated. The manuscript contains new interesting results in the field of coating. The addition of phosphorous can improve the magnetic properties of electrodeposited coatings. The authors explore the new sustainable electrolytic bath favoring the formation of Fe-W-P alloys with a higher content of P.

This work therefore deserves to be published in Coatings after having answered and inserted in the manuscript the comments of the questions below:

1-      Why the choice of the galvanostatic mode for the realization of the electrodeposition? The chronoamperometric or cyclic voltammetry modes can provide more interestingresults (to be commented).

2-      The Fe-W-P alloy in its amorphous state has magnetic properties. Is this the case for the same alloy in the crystalline state? Is it judicious to make a heat treatment to this alloy and to compare its magnetic properties with those of the amorphous state?

3-      Explain why you did not realize the effect of the other parameters on the electrodeposition of the Fe-W-P alloy? I site in particular the effect of the salt of font and the temperature of the electrolyte.

Author Response

Coatings-2309976-Electrodeposition of soft magnetic Fe-W-P alloy coatings from an acidic electrolyte.

In this paper, the authors studied the Fe-W-P coatings galvanostatic deposited from a newly developed electrolytic bath. The effect of plating parameters such as electrolyte current density and pH has been investigated. The manuscript contains new interesting results in the field of coating. The addition of phosphorous can improve the magnetic properties of electrodeposited coatings. The authors explore the new sustainable electrolytic bath favoring the formation of Fe-W-P alloys with a higher content of P.

This work therefore deserves to be published in Coatings after having answered and inserted in the manuscript the comments of the questions below:

  1. Why the choice of the galvanostatic mode for the realization of the electrodeposition? The chronoamperometric or cyclic voltammetry modes can provide more interesting results (to be commented).
  • Cyclic voltammetry and current efficiency are presented by quartz microbalance measurements. The galvanostatic mode is an industrial method of coating deposition and is the commonly used for the deposition of alloys (comp. literature).
  1. The Fe-W-P alloy in its amorphous state has magnetic properties. Is this the case for the same alloy in the crystalline state? Is it judicious to make a heat treatment to this alloy and to compare its magnetic properties with those of the amorphous state?
  • This is an interesting idea for future research, but in this paper we only consider amorphous alloys, which are also corrosion resistant coatings that can be used for applications such as MEMs, etc.
  1. Explain why you did not realize the effect of the other parameters on the electrodeposition of the Fe-W-Palloy? I site in particular the effect of the salt of font and the temperature of the electrolyte.
  • The advantage of this study is the demonstration of a new electrolytic bath with a new W source. The main focus was on studying the deposition mechanism and the influence of acidity, which is a major factor in the preparation of the bath. Previous studies have shown that lowering the temperature leads to an increase in the O content and a decrease in the P co-deposition in Fe-P alloys [1]. In addition, increasing the temperature above 60 °C leads to rapid evaporation of the bath, which makes it difficult to control the concentration in the bath. However, studying the temperature as well as the salt/acid content may be an attractive research for future.

[1]       N. Kovalska, N. Tsyntsaru, H. Cesiulis, A. Gebert, J. Fornell, E. Pellicer, J. Sort, W. Hansal, W. Kautek, Electrodeposition of nanocrystalline Fe-P coatings: influence of bath temperature and glycine concentration on their structural properties, mechanical and corrosion behaviour, Coatings. 9 (2019) 189. https://doi.org/doi:10.3390/coatings9030189.

Reviewer 4 Report

This manuscript considers the electrodeposition of an iron–tungsten–phosphorus alloy from an acidic electrolyte and characterizes a number of properties of the resulting coatings (in particular, magnetic properties). This interesting and useful manuscript deserves to be published in Coatings. I have only a few comments and questions about the work.

1. It is necessary to briefly describe the role of the main components of the electrolyte (Table 1) and at least try to justify the concentrations of the components used.

2. Justify the choice of the electrodeposition temperature (60°C). How does temperature affect current efficiency, alloy composition, and coating properties?

3. Is it possible to rewrite the conclusions on the article in a more concise and more clearly structured way?

Author Response

This manuscript considers the electrodeposition of an iron–tungsten–phosphorus alloy from an acidic electrolyte and characterizes a number of properties of the resulting coatings (in particular, magnetic properties). This interesting and useful manuscript deserves to be published in Coatings. I have only a few comments and questions about the work.

  1. It is necessary to briefly describe the role of the main components of the electrolyte (Table 1) and at least try to justify the concentrations of the components used.

Table 1. Bath composition.

Components

Molarity / M

Concentration / gl-1

FeSO4·7H2O

0.72

200

H2PO4·12WO3·H2O

0.02

60

Citrate

0.39

100

Glycine

0.64

48

NaH2PO2·H2O

0.08

7

(NH4)2SO4

0.45

60

  • FeSO47H2O, NaH2PO2·H2O, and Glycine was chosen according to the previous literature research and was described in papers [1–3].
  • H2PO412WO3·H2O acid was chosen due to its chemistry/composition. No literature is related to this.
  • (NH4)2SO4 was added for the stabilization.
  • Citrate supports the W co-deposition from the literature, the concentration was chosen due its stability. The lower concentration reduced life time of the bath. The bath was visually oxidized.
  1. Justify the choice of the electrodeposition temperature (60°C). How does temperature affect current efficiency, alloy composition, and coating properties?
  • The main focus was on studying the deposition mechanism and the influence of acidity, which is a major factor in the preparation of the bath. Previous studies have shown that lowering the temperature leads to an increase in the O content and a decrease in the P co-deposition in Fe-P alloys [1]. In addition, increasing the temperature above 60 °C leads to rapid evaporation of the bath, which makes it difficult to control the concentration in the bath. However, studying the temperature as well as the salt/acid content may be a good research for the future.
  1. Is it possible to rewrite the conclusions on the article in a more concise and more clearly structured way?
  • The conclusions are arranged in the structure as the paper is written. Starting from the QMB and finishing with Magnetic properties.

[1]       N. Kovalska, N. Tsyntsaru, H. Cesiulis, A. Gebert, J. Fornell, E. Pellicer, J. Sort, W. Hansal, W. Kautek, Electrodeposition of nanocrystalline Fe-P coatings: influence of bath temperature and glycine concentration on their structural properties, mechanical and corrosion behaviour, Coatings. 9 (2019) 189. https://doi.org/doi:10.3390/coatings9030189.

[2]       N. Kovalska, M. Pfaffeneder-Kmen, N. Tsyntsaru, R. Mann, H. Cesiulis, W. Hansal, W. Kautek, The role of glycine in the iron-phosphorous alloy electrodeposition, Electrochim. Acta. 309 (2019) 450–459.

[3]        N. Kovalska, W.E.G. Hansal, N. Tsyntsaru, H. Cesiulis, A. Gebert, W. Kautek, Electrodeposition and corrosion behaviour of nanocrystalline Fe–P coatings, Trans. IMF. 97 (2019) 89–94. https://doi.org/10.1080/00202967.2019.1578130.

Reviewer 5 Report

1.       Summary, strengths, weaknesses, overall contribution           

Summary: In the paper the Authors investigated Fe-W-P coatings deposited from a newly developed electrolytic bath. They have obtained amorphous structure with soft ferromagnetic properties.

General strengths: The interesting, modern and important topic.  

General weaknesses: The coatings should be studied in more details.

2.                   Major comments

From the electrochemical point of view, the paper is correct. Unfortunately, there are many standard properties of the coatings which are not taken into consideration in this work. For example, the surface topography, i.e. roughness should be studied. Furthermore, in Fig. 3 the surface is shown but there are still a lot of questions i.e. if the cracks are over all the surface or not. What was the influence of current density?

The Authors should also provide the cross-section of the coatings. What is the porosity of the coatings?

To interest wider audience at least basic mechanical properties should be provided. The Authors should measure hardness of the coatings. If possible, the internal stresses and their dependence on pH should be provided.

Author Response

Comments and Suggestions for Authors

  1. Summary, strengths, weaknesses, overall contribution           

Summary: In the paper the Authors investigated Fe-W-P coatings deposited from a newly developed electrolytic bath. They have obtained amorphous structure with soft ferromagnetic properties.

General strengths: The interesting, modern and important topic.  

General weaknesses: The coatings should be studied in more details.

  1. Major comments

From the electrochemical point of view, the paper is correct. Unfortunately, there are many standard properties of the coatings which are not taken into consideration in this work. For example, the surface topography, i.e. roughness should be studied. Furthermore, in Fig. 3 the surface is shown but there are still a lot of questions i.e. if the cracks are over all the surface or not.

  • The major goal of this paper was to study fundamental mechanism of the electrodeposition of new bath with new components. Also, the part of the bath composition control, and its influence on the alloy deposition. The surface topography, such as roughness and inhomogeneity distributed over the entire surface (bumps etc.) has been evaluated extensively supported by Fig. 3.

What was the influence of current density? Can you please specify?

  • The influence of current density is a major issue of chapter 3.2.

The Authors should also provide the cross-section of the coatings. What is the porosity of the coatings?

  • There were no porosities, except cracks at pH 3 deposition conditions, observed by high resolution SEM. Therefore, further cross-section were deemed not necessary in this study.

To interest wider audience at least basic mechanical properties should be provided. The Authors should measure hardness of the coatings. If possible, the internal stresses and their dependence on pH should be provided.

  • Mechanical properties were not of direct interest in the present study of the soft magnetic layers and their respective properties. A further study may focus on other aspects such as hardness.

Round 2

Reviewer 1 Report

Line 54, what makes this electrolyte sustainable? How does it differentiate from the state of the art? The answers to these questions should be clearly indicated.

Corrected: (Line 54-59)

Considering the numerous advantages of Fe-P coatings with high P content [9–11], the aim of this work is to explore a new sustainable electrolytic bath using non-hazardous materials, which leads to an increase in the P content in Fe-W-P alloys. Previous studies reported the formation of alloys, where the P content dominates over W. Therefore, we describe the use of a new source of elements and the influence of the chemical environment on the final alloy composition.

You should specify which hazardous materials you are not using in your approach, in opposition to the literature.

Line 65, how was the gold film deposited? It should be indicated.

Comment: We did not deposit the gold coating, it was ordered together with the equipment.

Ok.

Line 75, what steel was used? It should also be indicated.

Corrected (Line 80):

As counter electrode served a steel plate (type 304) exposing ~ 30 cm² to the electrolyte.

Formally, you should indicate the designation system such as AISI-SAE 304 or simply SAE 304.

Line 90, do authors mean ‘base’ instead of ‘basis’?

Corrected: (Line 95):

The conjugate base of this heteropoly acid is the oxyanion PW12O403−, where W has an oxidation state of +6.

Ok.

Line 107, I suggest presenting a table containing the semi-quantitative EDX readings of the films obtained from both electrolytes (with and without hypophosphate).

Corrected: Line 112 (Some numbers were added)

An EDX analysis of a deposit from an electrolyte without hypophosphate (NaH2PO2·H2O) resulted in practically zero (0.2 - 0.3 at.%) P content even though phosphate was present in the form of H2PO4·12WO3·H2O (comp. Table 1). In this case, the W content in the coating indicated 6 – 8 at. %.

Figure 1, how was charge density value obtained? You do not refer to this parameter anywhere else in the manuscript. What is its relevance and what information can be obtained from it?

Corrected: Line 118-119 (the value was obtained by QMB.)

A negative potential scan in the Fe-W and the Fe-W-P baths at pH 3 resulted in a negative current wave at ca. -0.4 V (Figure 1a) which can be interpreted as hydrogen evolution by the measured charge density |∆q| [9]. The absence of any frequency change (Df), i.e. any mass increase, indicates the generation of a gas uncoupled from the surface.

Ok.

Lines 133-134, did you consider brass substrate dissolution? Any comments on that?

Commented: Line 140

Since  a QMB with gold electrodes was used, no dissolution was observed

Ok.

Line 150, it should be said that this is only valid for pH = 4. Figure 2b clearly shows that at pH = 3 the current efficiency decreases with the current density.

Corrected: Line 155

The current efficiency (η) is higher in a less acidic electrolyte, i.e. at pH 4 and mostly unaffected by the change in current density (Figure 2b). This can be explained by a lower hydrogen evolution competing with the alloy deposition (comp. Figure 1b). In case of an electrolytic bath at pH 3, The current efficiency increases with higher current density [32] . The lowest current efficiency is observed at a current density of 30 mA cm-2 and pH 3 (Figure 2b) resulting in a slow deposition rate of the alloy and a maximum P content of ca. 13 at.% (Figure 2a).

Ok.

Figure 2, are these j values cathodic? If so, they should formally be negative or they should be explicitly indicated as cathodic.

Corrected: Figure 2

Line 167, commenting on the resolution, perhaps you should indicate the operation pressure inside SEM chamber, as well as beam energy.

What was the operation pressure and the beam energy inside the SEM chamber?

Lines 168-169, it is not clear to me how decreasing the conductivity of the phase, would result, per se, in lower resistance to corrosion/oxidation.

Corrected: The good resolution of the SEM images at all magnifications indicates low charging suggesting a high conductivity of the coating with an obviously low oxidic conversion overlayer.

Ok.

Line 170, how do you know the bumps are hollow? How do they compare to those obtained at pH 4? Perhaps some of the H2bubbles at pH 3 are just smaller and remain under the film.

Corrected: At pH4, is relatively high and practically independent of j. The H2-bubble formation and detachment are relatively slow, so that an overgrowth of the rapidly depositing Fe-W-P alloy leads to partially cracked film bumps which appear hollow (Figure 3)..

Ok.

Line 173, it is still not clear to me this correlation. Perhaps you could explain a bit better.

??

It was not clear what you mean, but you have already addressed this question in a previous point.

Figure 3, line 177, two pH and two current densities sound more adequate than ‘various’ in my opinion.

Corrected: Figure 3. Line 186

Ok.

Figure 4, for easier interpretation and formal representation, the xx axis should be arranged to include only the values of the classes considered: 30, 40, 50, and 60 mA/cm2.

The xx axis in Figure 4 should be redesigned to include only the current density classes used: 30, 40, 50, and 60 mA/cm2.

Line 210, why do you consider tungsten as a non-metallic element? Tungsten is a transition metal.

Corrected: Line 220 „The obtained results reveal that the Fe-W-P alloys are soft magnetic with coercively values lower than 25 Oe, although the percentages of the non-metallic element P in the alloy compositions were lower than 20 at.%.”

Ok.

Lines 219-220, it is stated that increasing P content increases coercivity. However, this seems to contradict what is suggested in lines 202-203, that the addition of W and P decreases coercivity. Also, I believe there is a typo in line 219 in ‘coercivity’.

Comment: The text above, it is said that addition of P and W reduces MS, but it does not say that it also reduces Hc.

Ok.

Line 241, where were these values (20 and 40 emu/g) read/obtained from?

Comment: The raw data was not normalized (the units were "emu"). It is common to present data divided by mass. Therefore the signal (in emu) was divided by the mass (in g) to obtain emu/g.

References list looks a bit outdated. The most recent papers are from 2019 (only 3/37) and most of the cited works are older than 10 years (21/37). I seriously recommend an update on the references.

Comment: Updated

Ok.

Author Response

Dear reviewer, 

Thank you for your contribution to improving this document. 

Below are selected amendments and comments:

Line 54, what makes this electrolyte sustainable? How does it differentiate from the state of the art? The answers to these questions should be clearly indicated.

Corrected: (Line 54-59)

Considering the numerous advantages of Fe-P coatings with high P content [9–11], the aim of this work is to explore a new sustainable electrolytic bath using non-hazardous materials, which leads to an increase in the P content in Fe-W-P alloys. Previous studies reported the formation of alloys, where the P content dominates over W. Therefore, we describe the use of a new source of elements and the influence of the chemical environment on the final alloy composition.

You should specify which hazardous materials you are not using in your approach, in opposition to the literature.

Corrected: In fact, after reviewing the technical data sheet of the acid we used, it turned out that it was not an environmentally safe chemical. So I corrected it accordingly.

Considering the numerous advantages of Fe-P coatings with high P content [9–11], the aim of this work is to explore a new electrolytic bath using sustainable or low quantity hazardous materials, which leads to an increase in the P content in Fe-W-P alloys. Previous studies reported the formation of alloys, where the P content dominates over W. Therefore, we describe the use of a new source of elements and the influence of the chemical environment on the final alloy composition.

Line 75, what steel was used? It should also be indicated.

Corrected (Line 80):

As counter electrode served a steel plate (type 304) exposing ~ 30 cm² to the electrolyte.

Formally, you should indicate the designation system such as AISI-SAE 304 or simply SAE 304.

Corrected.

Line 167, commenting on the resolution, perhaps you should indicate the operation pressure inside SEM chamber, as well as beam energy.

What was the operation pressure and the beam energy inside the SEM chamber?

Corrected:

“An acceleration voltage of 5 kV was used for the imaging mode, a voltage of 20 kV for the EDX mode. The vacuum pressure in the specimen chamber was of the order of 10-4 Pa.”

Line 173, it is still not clear to me this correlation. Perhaps you could explain a bit better.

??

It was not clear what you mean, but you have already addressed this question in a previous point.

Ok.

Figure 4, for easier interpretation and formal representation, the xx axis should be arranged to include only the values of the classes considered: 30, 40, 50, and 60 mA/cm2.

The xx axis in Figure 4 should be redesigned to include only the current density classes used: 30, 40, 50, and 60 mA/cm2.

Corrected.

Reviewer 3 Report

Dear authors

thank you for the comments.

You have done interesting work in the field of electrochemical layer deposition for various industrial applications.

In your next work, I suggest that you deal with all the parameters influencing electrodeposition.

Cordially

Author Response

Dear reviewer, 

Thank you for your contribution to improving this article. 

Best regards, 

corresponding authors

Reviewer 5 Report

The Authors have not modified the paper according to the reviewer comments. If the new bath is studied then at least basic information about the microstrucutre of the deposited coatings should be provided. Cross-sections, and XRD are necessary for different pH and currents densities to evaluate their influence.

Even the magnietic properties, which are the main interest for the Authors, are studied only for two samples.

Mechanical properties are not necessary but would also be very useful. Hardness measurements are not very challenging and time-consuming.

To sum up, I strongly recommend to reject the paper and allow the Authors resubmit it after significant improvements.

Author Response

Dear reviewer, 

Thank you for your contribution to improving this article. 

Comments for the reviewer below:

  • The Authors have not modified the paper according to the reviewer comments. If the new bath is studied then at least basic information about the microstrucutre of the deposited coatings should be provided. Cross-sections, and XRD are necessary for different pH and currents densities to evaluate their influence.

Comment: According to the literature and our previous work, the structure of this type of alloy is influenced by the alloy content. For example, we can observe the transition from the amorphous to the nanocrystalline state with increasing current strength, which directly affects the P content (decreases) and increases the grain size. We can observe a similar behavior when applying various pHs. This is a common observation that is described in the cited articles.

The purpose of this article is to study the mechanism of electrodeposition and to demonstrate how it influences the alloy content, not the structural analysis. There are no available studies that reveal such an alloy content. Therefore, the main message was to show the effect on the composition, i.e. the relationship between pH and hydrogen evolution.

  • Even the magnietic properties, which are the main interest for the Authors, are studied only for two samples.

Comment: All samples showed similar magnetic properties. Therefore, we have included only representative results from a few of them to see trends, hysteresis loops for 13% and 8% P content. The small deviation between the content of different alloys would be of interest for possible industrialization issues.

  • Mechanical properties are not necessary but would also be very useful. Hardness measurements are not very challenging and time-consuming.

Comment: Measurements of mechanical properties and hardness will certainly be of interest in future research. The focus of the present study was on fundamental mechanistic electrochemical issues. This is also expressed in the last sentence of the Introduction passage:

“Therefore, we describe the use of a new source of elements and the influence of the chemical environment on the final alloy composition.”

Best regards, 

corresponding authors